# Conceptual Framework of a Simulation-Based Manpower Planning Method for Construction Enterprises

Sławomir Biruk [1], Piotr Jaśkowski [1] and Magdalena Maciaszczyk [2,*]

1 Faculty of Civil Engineering and Architecture, Lublin University of Technology, 20-618 Lublin, Poland; s.biruk@pollub.pl (S.B.); p.jaskowski@pollub.pl (P.J.)

2 Department of Strategy and Business Planning, Faculty of Management, Lublin University of Technology, 20-618 Lublin, Poland

* Correspondence: m.maciaszczyk@pollub.pl

**Abstract:** The authors put forward a concept of a method to support workforce demand planning intended for construction contractors. The construction business suffers from shortages of skilled labor. The method facilitates analyzing the possibilities of project-to project resource sharing and uncertainties in the flow of new orders, both characteristics of construction. Construction contractors' manpower planning is based on a production plan. A precise and deterministic plan is possible only for a short planning horizon covering a fixed set of acquired orders. New orders are won irregularly, and the type and quantity of work they involve is uncertain. For this reason, the authors argue for using a simulation model of the company's production plan. Such a plan facilitates mapping the variability in the number of orders, their start dates and delivery times, as well as workloads. Simulating the flow of prospective orders helps assess the degree of utilization of the enterprise's human resources and the influence of the size of employment on the timeliness of the execution of individual projects. This provides a basis for determining reasonable employment levels. The approach helps determine the demand for manpower and provides the data necessary in the process of staff development and recruitment. Moreover, it provides a tool for assessing the feasibility of using in-house resources to fulfill new orders and the need to hire subcontractors. The novelty presented in the article is the method of manpower planning in construction companies, making it possible to take into account risk conditions in acquiring new orders and the variability of construction process execution times due to factors disturbing the work flow. The concept of the method was the basis for the development of a simulation model of the construction company's order execution under random conditions, allowing for the evaluation of the effects of maintaining different levels of employment of the workforce in terms of the degree of utilization of the production potential and timeliness of individual projects. Thus, the result of the research is a tool supporting the decision making of construction managers in human resources management practice.

**Keywords:** human resource management; manpower plans; simulation technique; production planning; risk

## 1. Introduction

Every enterprise seeks opportunities to grow and strives to gain competitive advantage through the efficient use of scarce resources. Human resource management implies investment in human capital [1]. Construction companies have mastered the quantitative aspects of short-term employment policy. However, in the face of the skilled labor scarcity observed in developed economies, rational employment policy must be long-term and focused on qualitative aspects. Sustainable human resource management should therefore be understood as a strategically coherent, comprehensive effort to direct and develop human resources within the structure of an organization [2–4].

People are not literally a resource, but they are disposed of resources of potential use for the enterprise: they possess qualities that enable them to perform useful roles in organizations. The components of human resources understood this way include knowledge, abilities, skills, attitude, values, and motivation [5,6].

Human resource management in a construction company is a continuous process that comprises the following stages:

- Setting the strategy by defining goals and the ways to achieve them, as well as policies for the personnel acquisition, development, and control in a long-term vision of the desired state and the associated system of tactical actions. The latter are expected to ensure the best use of human resources to achieve the long-term objectives of the organization;
- Manpower planning, fed with input from continuous monitoring of the organization's workforce requirements and anticipation of their changes;
- Recruitment and retention of human resources, taking actions to balance the demand for resources with access to a skilled workforce;
- Dismissing employees [7].

Understaffing is likely to result of the inability to obtain new orders (lost opportunity) and client dissatisfaction (delays and quality problems in orders already acquired) [8]. Overstaffing leads to non-productive costs [9]. Human resource planning needs to respond to the current and future personnel needs of the company. Employment planning is the starting point for the acquisition of human resources, which is a process that naturally takes time. It includes the analysis of the resources in hand, forecasting demand, and forecasting the supply of workers with needed qualifications. The uncertainties of such forecasting are related to both the prospects for new construction orders and development of the labor market.

Demand forecasting begins with an assessment of trends in the existing use of human resources in terms of the number of employees and their added value, which ensure the company functions properly for growth and innovation potential; the organization's outlook for procurement; and overall macroeconomic trends [10]. These are the basis for estimating the demand for workers with the required professional skills. Supply forecasting includes both analyses of the in-house resources and external (labor market) analyses. By juxtaposing projected demand with supply, management must ensure business security by planning to recruit or lay off employees, invest in retraining the workforce, or hire subcontractors [11].

To balance the labor supply with demand and make the most use of the resources at hand, it is necessary to consider career development and the upgrading of existing human resources. This means identifying the abilities, skills, and experience of employees, as well as identifying gaps in the demand for manpower that will prevent the organization from achieving its strategic objectives.

The aim of the paper is to develop the concept of a method supporting workforce planning in a construction company, taking into account the specificity of activity in the construction industry. Due to the fact that companies usually participate in the implementation of many orders at the same time, the proposed method not only must be based on the classically used techniques of work standardization, which can be applied to determine the workload in the implementation of individual processes and projects, but must take into account the risk and uncertainty in obtaining orders, and thus the variability of the scope of works and their quantity.

## 2. Theoretical View

### 2.1. Manpower Planning Methods

Proper management of human resources makes it possible to achieve the objectives of the enterprise and the construction company by hiring competent personnel and making full use of their skills, as well as creating proper working relations. Planning, a key element of human resource management, aims to adjust the number of employees with

appropriate skills to the requirements of the production plan, in accordance with the adopted development strategy, to ensure meeting the organization's objectives while reducing employment costs.

In assessing the demand for human resources, various techniques of quantitative analysis are used. The most common methods of quantitative analysis are as follows:

1. Expert judgement. This relies on the experience-based estimates furnished by team managers. They prompt the minimal and the optimal numbers of employees in their teams and list their initial qualifications and competencies. A modification of this technique is the Delphi method, which consists of collecting opinions of a group of experts and finding staffing solutions accepted by the majority of them.

2. Trend analysis. This is used to predict the future human resource demand on the basis of trends discovered in historical data. It is an assessment of changes on the internal and external labor market. Such analysis is more suitable for estimating personnel changes within teams of physical workers. It is reliable only in times of a relatively stable labor market.

3. Examining and standardizing work. This technique involves the analysis of tasks and the human resources required to perform them in a particular time. It is usually used when analyzing manual labor. It is often combined with other techniques.

4. Work process modelling. This involves the use of econometric techniques: operations research, statistics, and simulation to construct models of labor requirements and estimation based on econometric and statistical data.

5. Indicator method. Various types of indicators are used for planning the size or structure of employment, often determined on the basis of statistical data, e.g., determining the number of production workers per one administrative employee.

It should be noted that the basis for the application of these methods is the forecast of the volume of production and the scope of orders carried out in the future.

### 2.2. Manpower Planning and the Construction Enterprise

The basic activity of a construction contractor is to deliver works or complete built facilities in construction sites. From the point of a construction company manager, each construction is considered a project of a specific and finite duration defined by an as-planned start and completion date and a specific budget. A construction project is a set of coherent activities, including construction, assembly or renovation, testing, and commissioning of built facilities [12]. The construction projects are subject to risks and uncertainties [13–19]. They are carried out in a finite period of time using different procurement systems, and their implementation may involve many organizations, including not only construction companies, but also product suppliers, designers, banks, authorities, etc. Unlike projects, construction companies operate in the market continuously, adapting their production potential to the changing requirements and conditions of the environment, including the labor market.

As a rule, the enterprise, acting as the main contractor or subcontractor, participates in several construction projects at the same time. The number of projects varies over time. Operational planning of particular projects relies on the almost complete information on the scope of works; thus, the scheduling and budgeting of individual construction processes is a rather routine operation. However, planning a production program for the entire enterprise is a continuous process, with a rolling horizon and an uncertain scope of projects involving those that are anticipated but not won yet. New projects appearing in the enterprise's portfolio may be considered independent from the projects in progress. However, as the projects share at least some part of the enterprise's resources, technological and organizational dependencies between them are inevitable and need to be accounted for in the planning process.

The problem of human resource management, including workforce planning, on an enterprise scale therefore requires taking into account both the production needs resulting from the implemented projects and the availability of the labor force in the labor market. It

is therefore important to develop methods of forecasting employment on the scales of the domestic market, the project, and the company.

The human resource pool, defined by the number and the skills of the crew, determines the enterprise's production capacities. The analysis of in-house resources in terms of their condition, availability, and allocation to particular tasks, as well as the necessity of obtaining new resources from external sources, is carried out at each level of enterprise and construction project management. The goal of construction enterprise management is to make the most of in-house resources, but with uncertainties related with the acquisition of new orders in a highly competitive market, it proves difficult to balance the production capacity of the enterprise and the actual demand [20,21]. Bidding strategies to win "the right number of orders" [22] are considered a way to achieve this balance.

The efficiency of construction enterprises can be increased by eliminating resource downtime, adjusting the number of resources to the available work space, and leveling resources within a particular project as well as from the enterprise perspective. Optimizing the enterprise resource plans "per project" does not guarantee the optimal resource management in the enterprise as a whole. The accuracy of forecasts in determining the number and size of future orders decreases with the extension of the planning horizon. Even with a fixed set of orders of the company, the execution of individual projects is affected by risk and uncertainties that affect in-house resource efficiency.

The workforce structure, especially in terms of the number and qualifications of the workers, is not uniform in all construction companies. It depends on the size of the enterprise, its specialization, the territorial range, and the levels of mechanization of the works. An efficient employment plan facilitates allocating resources and scheduling projects, provides the data necessary for recruitment and employment of workers, and contributes to the rational use of the personnel [23,24]. The availability of qualified workers affects the quality of work and the client satisfaction. Only a sufficient number of employees with appropriate qualifications makes it possible to efficiently allocate resources, reduce time, and cut costs.

As for the construction labor market as a source of new employees, there is rarely an equilibrium between the demand and the supply. Labor demand in the construction industry is cyclical and random, which results in a shortage or surplus of workers [25,26].

Proper HR planning affects the enterprise's capacity, increases the economic efficiency of ventures, achieves higher quality, and ensures personal development of employees [27] as well as personal security in its broadest sense [28]. HR planning is implemented in the strategic dimension (long term) and in the tactical and operational dimension (quarterly and/or monthly plans). The construction company is expected to develop a human resource management strategy that ensures the level of employment of workers with appropriate skills adapted to the production plan, with an indication of the sources of their acquisition (internal and external), creates conditions that encourage employees to remain in the organization, and sets directions for increasing productivity and efficiency. An insufficient number of employees may result in the inability to take new orders [23] and delays in the execution of projects in progress, which results in delay penalties or even losing contracts. Overstaffing generates non-productive costs and ultimately leads to losing the ability to compete.

Human resource planning in construction is an ongoing process that involves regular reviews of the organization's current resources and forecasting future needs based on the company's production plan. In forecasting the supply of personnel, the internal and external labor market must be considered. First, the internal market of the organization is analyzed by determining the state and structure of current employment. Then the external market is forecast, taking into account the unemployed, graduates, and employees employed in other organizations who are willing to change employers. Shortages of skilled workers in the market in the times of high demand may result in the enterprise relaxing the requirements, which may result in a lower quality of work, the need to work

overtime to complete construction processes on time, and wage increases to retain qualified resources [25].

Forecasting labor demand models allows for balancing manpower supply and demand. These models are mostly created at the national aggregate manpower level [29–31] and the project level [32–36].

Models developed at the national aggregate manpower level can be helpful in forecasting labor demand trends and formulating policies, training, and re-training programs as a response to changes in construction demand and demographic changes [37,38]. Many models have been developed to predict future employment levels at the national level. Early models are based on neoclassical economic theories. Current models are based on statistics, including linear and nonlinear regression models as well as univariate and multivariate time series models [25]. In statistical models, future employment demand is described by independent variables such as material price, construction output, labor productivity, real wage [31], and bank rate [29].

At the project level, simple relationships between labor demand and construction costs or productivity rates are most often modeled. Estimating labor demand by trade is usually ignored [37]. Manpower forecast models at the project level by trade can be directly applicable as inputs for scheduling construction projects using the critical path method, determining manpower requirements and resource leveling [32].

Bell and Brandenburg [32] developed a regression model expressing labor demand for various trades as a function of the project type and the labor cost. The input came from records of 130 completed highway construction projects. Chan et al. [33] presented a forecasting model that used labor multipliers calculated on the basis of the relationship between the level of employment and incurred expenditures in specific periods of the project's makespan. The multipliers were determined using data of 61 projects in Hong Kong. They enable the user to estimate the demand for 38 trades. Agarwal et al. [35] found that the factor of the greatest impact on employment levels was the project cost. The authors developed regression models for four categories of employees (engineers, consultants, administrators, and shop floor).

The above models have limited application. They allow forecasting the demand for manpower in a certain type of project, e.g., road projects or commercial building construction projects, as created using datasets on projects of a similar nature. They focused on selected trades and ignored changes occurring in construction methods [35]. For this reason, the model presented by [37] allows forecasting the demand for 10 trades using construction-specific independent variables affecting employment levels, such as the project cost, project complexity, site conditions, or project type. Moreover, the models described in the literature are applicable to certain local markets from which the data on completed construction projects were obtained [33].

The literature does not present methods of planning at the level of the entire enterprise, thus covering simultaneous implementation of a number of projects. So far, no tools have been developed to support the strategic management of human resources. However, the quality of management decisions at the tactical and operational planning level can be improved with dedicated optimization methods and computerized decision support systems [39–44].

## 3. Materials and Methods

A simulation model of the production plan of the enterprise is proposed to facilitate the analyses of the impact of resource availability on the duration as well as the delays of the start and finish dates of individual construction projects.

Analytical models of real phenomena are created to describe simple production systems. As a rule, they do not allow reflecting complex conditions of their functioning and relations between system elements and the environment. Simulation models do not have these limitations and make it possible to conduct experiments that are impossible to carry out in real conditions; therefore, they are often used to forecast and predict the behavior of

systems in the future. The research on the virtual model does not interfere in the system under study, but it can provide valuable information to support optimal decision making.

Simulation helps determine the required number of workers in each trade to assure timely delivery of projects with reasonable resource utilization rates. At the stage of strategic planning, this information is necessary to adjust the contractor's resources (level of employment, number of crews) to the prospective portfolio of projects. In addition, the simulation model allows the planner to determine the probability distribution of the project duration according to resource constraints that result not only from the assumed limits of their numbers but also from the concurrent implementation of construction processes at all construction sites. The distribution of the project duration can be the basis for determining the project dates at a particular level of reliability. The accuracy of such estimates is greater than the accuracy achievable by traditional methods that schedule each project separately.

The purpose of simulation studies may be, among others, to determine the optimal technical and organizational structure of the execution system, to determine the strategy of resource allocation, or to determine the executive potential of a construction company, ensuring the timely execution of orders. A proper executive structure of a construction company ensures elimination of the necessity to subcontract a part of the scope of works, which increases the company's profit and ensures higher quality and timeliness of work, as it is performed by its own proven working teams.

Simulation models have been used to describe, plan, and study complex systems in the construction industry for several decades [16]. By synthesizing the estimable random influences on the execution time of individual processes, simulation models allow the user to estimate the progress of the entire project and the execution of the production plan of the construction enterprise. The construction enterprise can be treated as a system operating theoretically without interruption, and the simulation model has a steady-state simulation character, i.e., the boundary conditions do not affect the states of the modeled system after the transition period.

Networks are used to model construction projects. Project scheduling methods based on the use of network models have been used in the construction industry for several decades. Using a network drawn up in the vertex convention (one-point network), the technological and organizational conditions of a construction project (sequence relations between construction processes) can be described using a unigraph $G = \langle V, E \rangle$ directed, coherent, and acyclic, with one initial and one final vertex, where $V = \{1, 2, \ldots, n\}$ is the set of construction processes (activities in the network) and $E \subset V \times V$ is a binary relation describing the sequence relationships between processes. This function $T : V \to R^+$ assigns execution time $t_i$ to processes $i \in V$. Thus, with the help of networks, it is possible to model the scope of construction works in progress, their execution time, and the dependencies between them. The analysis of network models allows determination of deadlines for process execution and creation of schedules and allocation of work resources for task execution.

The biggest disadvantage of classical network methods, such as the project evaluation and review technique (PERT) or critical path method (CPM) [45], used for scheduling is the assumption of the unlimited availability of resources [46]. In the process of scheduling, limited quantities of resource units must be taken into account, as they can affect the time of implementation of processes, the course of the critical path, and even the time of implementation of the entire project. From the point of view of the construction company, when allocating resources, it is necessary to take into account all simultaneous undertakings of the company, for the execution of which the same resources are engaged. When setting the dates for the commencement of processes, not only sequence constraints (the necessity of completing all preceding processes) resulting from the technology and the adopted concept of work organization but also constraints in the availability of shared resources must be taken into account.

## 4. Results and Discussion

The proposed concept is based on the assumption of variability of the scope of works due to new orders added to the project portfolio at random moments as well as the random character of the duration of construction processes. It includes the following stages:

I.  Collecting the input to develop a model of the company's production plan. The data needed for simulation modeling include:

- The types of ongoing construction projects (works) modeled by means of networks;
- The intensity of the arrivals (starting subsequent projects);
- The distribution types and parameters of process execution times;
- The list of key resources (their type and demand) to deliver the processes;
- The predefined limits on the availability of resources.

II. Building the simulation model of the enterprise's production plan. To model the enterprise's production plan, one needs to determine the types of prospective projects and the desired number of projects. For each type of project, a generic network model is prepared. For instance, the typical orders taken by one of the studied construction enterprises, specializing in housing construction in a local market, could be classified into only four project types:

Type 1: Erection of a single housing block of about 1000 m$^2$ of usable floor area (20 to 25 apartments).

Type 2: Erection of a single housing block of about 2200−2500 m$^2$ of usable floor area (50 apartments).

Type 3: Erection of a single housing block larger than 5000 m$^2$ of usable floor area (100 apartments).

Type 4: Erection of a set of housing blocks to be delivered in stages; most frequently, the sets comprise two similar blocks, each of about 2200−2500 m$^2$ of usable floor area.

The scope of work and logic of works are identical for each project type, so they can be modeled with identical graphs. Projects are modeled as arrivals of a queuing system.

In the simulator, it is necessary to define the probability distribution of intervals between the arrivals of the same type. In addition, it is required for each type of project to define the incidence matrix (sequence dependencies), the type and number of resources needed for the execution of each process, and estimates of the parameters of the distribution of the execution time of the processes. Additionally, the level of availability of key resources and their cost of use should be iteratively determined. For resources of lesser importance, it can be assumed that their availability is not limited.

A process, whose predecessors have been completed and whose resources are available in quantities necessary for its execution, is added to the set of processes ready to start. The selection of processes for execution (allocation of a limited number of resource units to the process) is made based on heuristic rules that determine the priority values assigned to tasks. These rules are based on the duration or on the completion date of the processes, on the process cost, or on delay penalties. The Shortest Processing Time, Longest Processing Time, Earliest/Latest Start Time, Most Total Successors, Most Immediate Successors, Most Critical Successors, Minimum Earliest/Late Finish Time, etc. can be selected as the rules [47–50].

A conceptual flowchart of the simulator is shown in Figures 1 and 2. The production plan simulator was programmed in the general purpose programming language Java Standard Edition 8 in the Eclipse IDE for Java Developers.

III. Model verification and validation. The verification of the model should be conducted at the stage of the model's computer implementation and as the model is completed to check if the model performs according to the conceptual description. Then, the model needs to be validated, i.e., checked to what degree it describes the real system [51].

Due to the one-time nature of construction projects, model validation is difficult. Most often, it involves comparing the generated results with the results of other models (analytical or verified simulation) or with the records of actual projects.

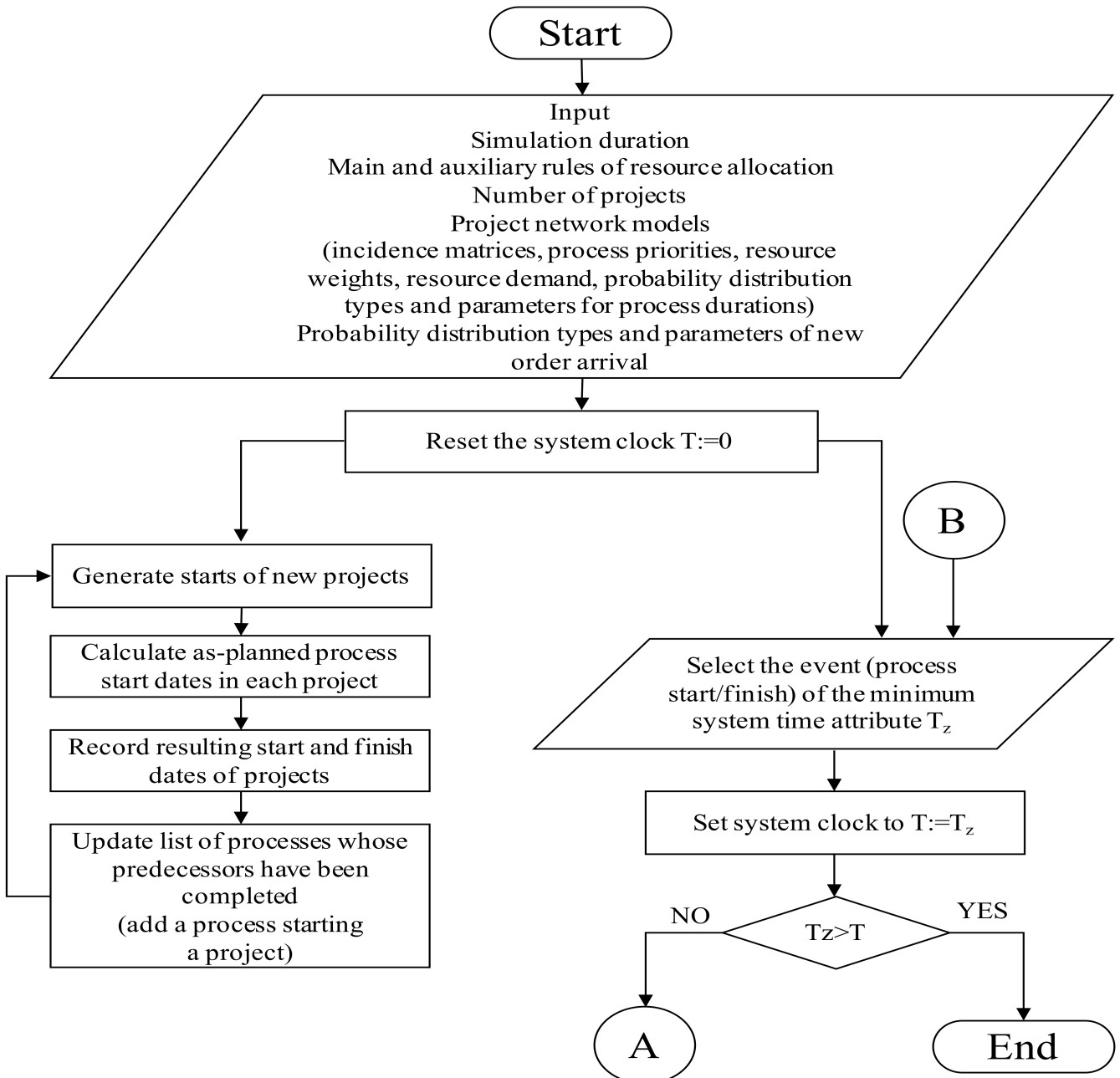

**Figure 1.** Flowchart of the simulation model of the contractor production plan. Source: own creation.

IV. Simulations. Simulation studies are designed to analyze the impact of input parameters (mainly the level of resource availability and the selected resource allocation heuristic rule) on the output parameters. Performance measures of the proposed solution should be consulted by the company's management and be in line with the company's strategy.

The system operation can be evaluated using a variety of indicators, such as

- the number of projects completed within a specified period;
- the average project duration;
- the average value of the extension of time;
- the number or cost of resource transfers from project to project;
- the resource utilization rates;
- the cost of resource idle time;
- the costs of delay penalties.

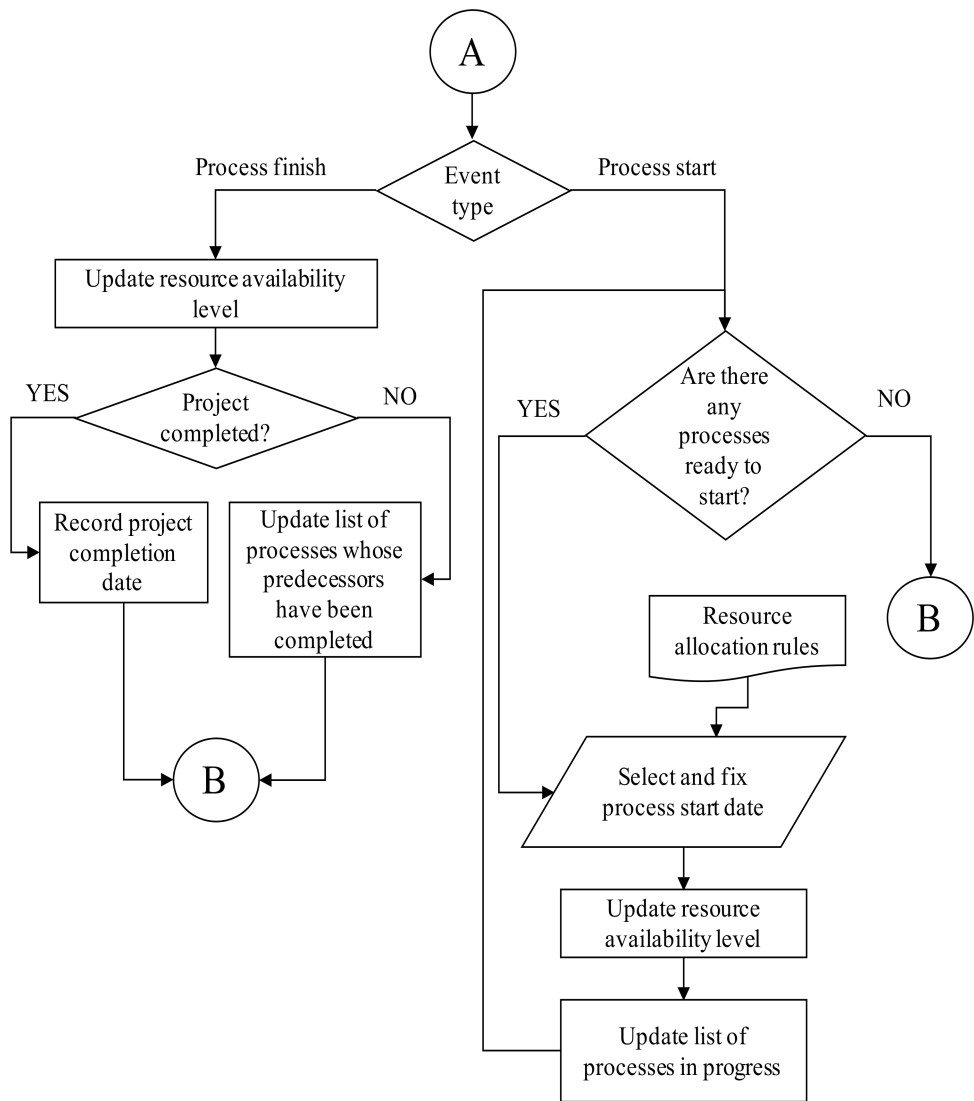

**Figure 2.** Flowchart of the simulation model of the contractor production plan (continued). Source: own creation.

They can form the basis for formulating criteria for evaluating different variants of the workforce plan. Their selection depends on the preferences of the decision maker (e.g., he or she may seek to maximize the average number of completed orders in a given period, minimize the average time of project implementation, or minimize the average time of extension of the implementation time beyond the directive deadline-considering individual criteria). A number of indicators can be considered at the same time by means of a multicriteria utility function. This, however, involves expressing the criteria in a common unit, typically a monetary unit. This facilitates finding the optimal solution in terms of the adopted utility function; however, the process of criteria standardization results in a partial loss of information.

Let us assume that the simulation study was conducted for m scenarios of workforce deployment plans $S_j \in \{S_1, S_2, \ldots, S_m\}$, differing in the number of workers of particular qualifications. As a result of the simulation experiments (representing one year of the company's production plan), the following results were obtained:

- $K_T(S_j)$—the cost of resource transfers from project to project for the scenario $S_j$,
- $K_I(S_j)$—the cost of resource idle time for the scenario $S_j$,
- $K_P(S_j)$—the costs of delay penalties for the scenario $S_j$.

The $S_j$ scenario, for which the combined cost of resource downtime, financial penalties for exceeding directive deadlines, and resource redeployment is minimal, should be considered optimal:

$$\min K(S_j) : K(S_j) = K_T(S_j) + K_I(S_j) + K_P(S_j) \tag{1}$$

It is recommended that simulation studies are conducted first under the assumption of unlimited resource availability. This enables the planner to estimate the shortest project durations and resource demand levels as well as their utilization rates. These system operation metrics may be used as benchmarks to evaluate solutions obtained during further analyses that account for resource limits. On this basis, after acceptance by the management, it is possible to set the ranges of resource variability in further simulation studies.

The analysis of output allows the planner to determine the level of employment of key resources and their impact on the project start and completion dates under random conditions.

V. Development of a monitoring system. Due to the natural changes in the company's environment, it is necessary to continuously monitor changes of the market and update the model in terms of the types of prospective orders and the distribution parameters of their acquisition (arrivals to the system). Updates may be needed due to changes in the enterprise's bidding strategy or decisions to enter new market segments. It is also necessary to analyze the possibility of increasing or decreasing the production potential of the enterprise by analyzing its financial standing. These actions must be in line with the company's strategy.

VI. Implementation. The numbers of units of the key human resource, the project durations, and differences between the as-planned and simulated completion dates can prompt the management's rationale for the levels of employment and equipment of the construction enterprise. The results of the analysis help justify the need to invest in or to reduce the production potential and introduce changes in the management system.

VII. Control. In the face of organizational changes and technological advances, it may be necessary to design new network models and re-estimate the construction process durations. Keeping records of completed construction processes and analyzing the reasons for discrepancies between the actual and the as-planned completion times may be helpful in this process. It may prove necessary to modify the simulation model and to account for more constraints, e.g., to consider the correlation between durations of processes affected by the same risks or factors, such as the season of the year. At this stage, practical validation of the developed method and simulation model is carried out. The analysis of the consistency of assumptions made in the construction of the model with the real values of the data mentioned in Point I allows modification of the model and an increase in the reliability of the obtained results.

## 5. Conclusions

Construction contractors and businesses that provide construction management services need to coordinate the work of many people and organizations in multiple construction sites. Timely completion of a particular project does not guarantee the success of the whole enterprise. The company strives to achieve the objectives of individual projects (orders), measured in terms of time, cost, and quality, using in-house resources and reliable subcontractors. The acquisition of resources on the market, especially in times of economic prosperity, becomes harder due to the scarcity of skilled labor. The risk of delays and inferior quality due to labor shortages is thus growing. Therefore, the contractor is forced to keep a sufficient number and type of resources to enable efficient execution of works; the subcontracting of works and acquiring seasonal workers should be limited to specialty works. However, keeping resources (acquiring, training, convincing to stay) must be economically justified; the resources must be well utilized.

Human resources are a key factor in the success of a company. The availability of manpower with the right skills and in the right quantity allows for timely completion of

orders. Construction contractors are subject to contractual penalties if they fail to meet directive deadlines. On the other hand, completing a project ahead of schedule is sometimes rewarded by the investor, who can start operating the constructed facilities earlier and shorten the period of return on invested capital. In addition, the company receives payment for completed work earlier (shorter period of freezing working capital in work in progress) and can apply for new orders. However, from the point of view of the enterprise, too high a level of employment is inefficient due to higher labor costs and losses connected with not using the production potential. Therefore, it is important to develop reliable methods to support enterprise-wide workforce planning to reduce the costs associated with maintaining a given level of human resource employment.

The proposed methodology for modeling the execution of a portfolio of construction orders helps determine the demand for in-house resources, to ensure timely execution of projects and keep utilization rates high. The developed method was the basis for the development of a simulation model for the implementation of the production plan of a construction company. The approach, based on simulating the execution of the production plan, enables the planner to consider construction-specific constraints: fluctuations of the workload due to adding new orders to the project portfolio, limited supply of labor, the need to transfer resources from one construction site to another, and above all, the cyclical character of planning and executing projects within a planning horizon specified by the decision-maker. The applied simulation approach allowed taking into account the impact of random conditions of construction enterprises on their effectiveness in terms of the degree of utilization of production potential and employed human resources, as well as the possibility of timely execution of concluded contracts. It seems that despite the uncertainty about the number and scope of contracts to be won in the future, as well as the impact of risk factors on the course of orders and individual construction processes, it is possible to reliably predict future production needs in terms of the size and structure of the workforce.

Application of the proposed method in practice requires acquisition and statistical analysis of data on past construction projects. Therefore, it is impossible to develop a universal simulation model that can be used in every enterprise; it is necessary to modify it individually and adapt it to the existing technical and organizational conditions.

Planning the execution of an evolving portfolio of orders is important for construction practice. The modeling of this process is very complex and requires further research. It involves focusing on the development and verification of statistical methods to forecast the arrivals of orders, which is particularly challenging, as the construction market is highly competitive. The future models for construction production planning are expected to combine the forecasts of the volume of work with the models of bidding strategies of a construction company.

**Author Contributions:** S.B., P.J. and M.M. confirm that their contributions in each stage of the preparation of this article were equal. All authors have read and agreed to the published version of the manuscript.

**Funding:** This work was funded under the grant "Subvention for Science" (MEiN), project no. FN-6, FN-10, FD-20/IL-4/005, FD-20/IL-4/026.

**Institutional Review Board Statement:** Not applicable.

**Informed Consent Statement:** Not applicable.

**Data Availability Statement:** Not applicable.

**Conflicts of Interest:** The authors declare no conflict of interest.

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
