# Peer review of "Conceptual Framework of a Simulation-Based Manpower Planning Method for Construction Enterprises"

_sustainability, doi:10.3390/su14095341_

Round 1
Reviewer 1 Report
In the abstract, the results and conclusions of research should be added. Please rewrite the abstract section.
Author Response
Dear Reviewer,
thank you very much for your valuable comments. The process of making revisions after reviews is not easy, but we did our best to meet expectations.
In accordance with the recommendation, the abstract was enriched with a discussion of the results obtained and a summary of the study. Now the abstract includes information that the concept of the method was the basis for the development of a simulation model of the construction company's order execution in random conditions, allowing for the evaluation of the effects of maintaining different levels of employment of the workforce in terms of the degree of utilization of the production potential and timeliness of individual projects. Thus, the result of the research is a tool supporting the decision-making of construction managers in human resources management practice.
We sincerely hope that our explanations meet with your approval and allow us to share the results of our research.
Authors
Reviewer 2 Report
The paper claims to propose a conceptual framework to support workforce demand planning. However, the framework is not clear as no equations or criteria to find workforce demand were written. The framework was not also validated, even no results obtained from the framework were presented. Therefore, the paper is not recommended for publication to ensure the quality of the journal.
Author Response
DóÅ‚ formularza
PoczÄ…tek formularza
Dear Reviewer,
thank you very much for your comments. The process of making revisions after reviews is not easy, but we did our best to meet expectations.
The indicator methods indicated in Stage IV can form the basis for formulating criteria for evaluating various variants of the workforce plan. Their selection depends on the preferences of the decision-maker (e.g. he may seek to maximize the average number of orders realized in a given period, minimize the average time of project realization, minimize the average time of extension of the realization time beyond the directive deadline - considering single criteria). A number of indicators can be considered at the same time by means of a multicriteria utility function.
The paper proposes such an objective function in Section 4, with the aim of reducing resource downtime costs, financial penalties for exceeding directive deadlines, and resource turnaround costs.
In the paper, due to volume limitations and the complexity of the models used in practice, it was not possible to present them graphically and discuss an example. Chapter 4 discusses only the types of objects that were included in the analysis conducted on the example of one of the enterprises - they were adopted on the basis of interviews with the management of the studied enterprise.
Practical verification of the developed simulation model is difficult due to the high cost intensity of construction production. It is also difficult to compare the results obtained with the results obtained with other methods - because the ones that are used in practice allow only to determine the need for manpower for the implementation of individual projects and do not take into account the simultaneous implementation of many projects and variability of the production plan.
Practical verification is part of stage VII of the proposed method: "At this stage, practical validation of the developed method and simulation model is conducted. The analysis of the consistency of the assumptions made in the construction of the model with the actual values of the data listed in section I allows modifying the model and increasing the reliability of the results obtained."
We sincerely hope that our explanations meet with your approval and allow us to share the results of our research.
Authors
Reviewer 3 Report
The assessed article can be evaluated positively, brings new knowledge and is a benefit for science and research.
However, I have two important comments.
I recommend expanding the list of used scientific sources with other scientific works that can contribute to higher scientific value:
- Peracek, T. (2020). Human resources and their renumeration: managerial and legal backround. 13th International Scientific Conference on Reproduction of Human Capital - Mutual Links and Connection (RELIK) 2020 | RELIK 2020: REPRODUCTION OF HUMAN CAPITAL - MUTUAL LINKS AND CONNECTIONS, pp.454-465
- Jankelová, N., Jankurová, A., Beňová, M. & Skorková, Z. (2018) Security of the business organizations as a result of the economic crisis. Entrepreneurship and Sustainability Issues, 5 (3), pp. 659–671, doi: 10.9770 / jesi.2018.5.3 (18)
- Srebalová, M. & Vojtech, F. (2021). SME Development in the Visegrad Area. Eurasian Studies in Business and Economics, 17, pp. 269–281, doi: 10.1007 / 978-3-030-65147-3_19
- Shevchuk, O .; Matyukhina, N; (...); Volianska, O. (2021). The human right to security in the implementation of the concept of the "right to health protection" JURIDICAL TRIBUNE-TRIBUNA JURIDICA 11 (3), pp.535-548
I recommend extending the conclusion of the article and more precisely defining the results obtained as well as the answers to the established hypotheses / research questions.
Author Response
DóÅ‚ formularza
PoczÄ…tek formularza
Dear Reviewer,
We sincerely thank you for recognizing the benefits of the article for science and research. In accordance with the reviewer's suggestion, the bibliography was expanded to include the indicated positions, which enriched the article with a new perspective on the discussed issues.
Obtained results were indicated in the abstract, as well as discussed in the conclusion section, pointing to the problem of decision-making under risk and uncertainty in the activity of construction companies. We sincerely hope that our explanations meet with your approval and allow us to share the results of our research.
Authors
Reviewer 4 Report
This manuscript provided a method to plan the human resources in construction enterprises using simulation models. The overall structure is complete and logical. However, there are some questions that need to be addressed.
1. Section 2.2: Please explain what is “the level of the entire enterprise” and its difference compared with national and project levels.
2. Please explain the significance of proposing methods of planning at the level of the entire prise.
3. What is the research problem that the proposed method aims to solve?
4. Section 3: Please explain why the simulation model is selected as the method rather than the other reviewed methods.
5. Why networks are used? What are the networks here?
6. The explanation of the development process and methodology of the networks are too simple without clear analysis and valid support.
7. Section 4: Why these data are selected? Please provide the basis.
8. How do you validate the proposed method? Or how to demonstrate that the proposed method is effective and reasonable?
9. Section 5: Please state the limitations of this study.
Author Response
Dear Reviewer,
thank you very much for your valuable comments. The process of making revisions after reviews is not easy, but we did our best to meet expectations.
- In Section 2.2: We have clarified the national construction industry's project-specific considerations and the possibility of different companies participating in the projects, as well as the simultaneous implementation of multiple projects by single companies. A paragraph was added: “The problem of human resource management, including workforce planning, at the enterprise scale, therefore, requires taking into account both the production needs resulting from the implemented projects, as well as the availability of labor in the labor market. It is therefore important both to develop methods of forecasting employment both in the scale of the domestic market, the project, as well as the enterprise”.
- The significance of the proposed method has been further described in conclusion section: "Human resources are a key factor in the success of a company. The availability of manpower with the right skills and in the right quantity allows for timely completion of orders. If directive deadlines are not met, construction contractors are subject to contractual penalties. On the other hand, completion of the project ahead of schedule is sometimes rewarded by the investor, who can start operating the constructed facilities earlier and shorten the payback period of the invested capital. Also, the company earlier receives payment for completed works (shorter period of freezing working capital in work in progress) and can apply for new orders. However, from the enterprise's point of view, overstaffing is inefficient due to higher labor costs and losses due to unused production potential. It is therefore important to develop reliable methods to assist in enterprise-wide workforce planning to reduce the costs associated with maintaining a given level of human resource employment."
- The purpose of the article and the research problem have been indicated in the introduction section: "The aim of the article is to develop a concept of a method supporting workforce planning in a construction enterprise taking into account the specificity of activity in the construction industry. Due to the fact that companies usually participate in the implementation of many orders at the same time, the proposed method must be based not only on the classically used techniques of work standardization, which can be applied to determine the workload in the implementation of individual processes and projects, but must take into account the risk and uncertainty in obtaining orders, thus the variability of the scope of works carried out and their quantity.
- In section 3, the choice of research tools used has been justified: "Analytical models of real phenomena are created to describe simple production systems. As a rule they do not allow reflecting complex conditions of their functioning and relations between system elements and environment. Simulation models do not have these limitations and allow conducting experiments that are not possible to carry out in real conditions, therefore they are often used to forecast and predict the behavior of systems in the future. Testing of virtual model does not interfere in the studied system and it can provide a lot of valuable information supporting optimal decision making."
- Regarding networks - project scheduling methods based on network models have been used in the construction industry for several decades. Networks can be used to model the scope of a construction project, the timing of its execution, and the relationships between them. The analysis of network models makes it possible to set deadlines for process execution and to create schedules and allocate labor resources for task execution. (This has been further explained in part 3 of the article).
- In this paper, due to volume limitations and the complexity of the models used in practice, it was not possible to present them graphically and discuss an example. Chapter 4 discusses only the types of objects that were included in the analysis conducted on the example of one of the companies.
- The data mentioned in Chapter 4 were necessary for the construction of network models of the realized types of projects and for the construction of the network model of the production plan of the construction enterprise. In the proposed conception of the method, it was assumed on the basis of interviews with the management of the studied enterprise that its production plan includes the realization of 4 types of objects. The development of network models of their construction projects required the acquisition of additional data on the execution times of individual processes and on the frequency of obtaining particular types of orders. The simulation of the course of the realization of the projects allowed for a dynamic allocation of human resources for the execution of the processes, taking into account the initially assumed restrictions (changed during the research).
- Practical verification of the developed simulation model is difficult due to the high cost-intensity of construction production. It is also difficult to relate/compare the obtained results to the results obtained by other methods - because those that are used in practice only allow to determine the demand for manpower for the implementation of single projects and do not take into account the simultaneous implementation of many projects and variability of the production plan. Practical verification is part of stage VII of the proposed method: "At this stage, practical validation of the developed method and simulation model is conducted. The analysis of the consistency of the assumptions made in the construction of the model with the actual values of the data listed in section I allows modifying the model and increasing the reliability of the results obtained."
- In Section 5, the following weaknesses of the proposed approach have been identified: "The application of the proposed method in practice requires the acquisition and statistical analysis of data on past construction projects. Therefore, it is not possible to develop a universal simulation model that can be applied to every enterprise - it is necessary to modify it individually and adapt it to the existing technical and organizational conditions.”
We sincerely hope that our explanations meet with your approval and allow us to share the results of our research.
Authors
Round 2
Reviewer 4 Report
The authors have clearly addressed the comments.